# Xanthate-Modified Magnetic Fe_3_O_4_@SiO_2_-Based Polyvinyl Alcohol/Chitosan Composite Material for Efficient Removal of Heavy Metal Ions from Water

**DOI:** 10.3390/polym14061107

**Published:** 2022-03-10

**Authors:** Shifan Wang, Yuan Liu, Aiwen Yang, Qi Zhu, Hua Sun, Po Sun, Bing Yao, Yunxiao Zang, Xihua Du, Liming Dong

**Affiliations:** 1School of Material and Chemistry Engineering, Xuzhou University of Technology, Xuzhou 221018, China; shifanwang@xzit.edu.cn (S.W.); ly17305187957@163.com (Y.L.); yaw8023wbh@163.com (A.Y.); zq010214@163.com (Q.Z.); iamsunhua@xzit.edu.cn (H.S.); yaoming@xzit.edu.cn (B.Y.); yxzang@zxit.edu.cn (Y.Z.); dxh@xzit.edu.cn (X.D.); 2Analysis and Testing Central Facility, Anhui University of Technology, Maanshan 243032, China; sunpoo@ahut.edu.cn

**Keywords:** magnetic materials, chitosan, adsorption, heavy metal, modification

## Abstract

Chitosan has several shortcomings that limit its practical application for the adsorption of heavy metals: mechanical instability, a challenging separation and recovery process, and low equilibrium capacity. This study describes the synthesis of a magnetic xanthate-modified polyvinyl alcohol and chitosan composite (XMPC) for the efficient removal and recovery of heavy metal ions from aqueous solutions. The XMPC was synthesized from polyvinyl alcohol, chitosan, and magnetic Fe_3_O_4_@SiO_2_ nanoparticles. The XMPC was characterized, and its adsorption performance in removing heavy metal ions was studied under different experimental conditions. The adsorption kinetics fit a pseudo-second-order kinetic model well. This showed that the adsorption of heavy metal ions by the XMPC is a chemical adsorption and is affected by intra-particle diffusion. The equilibrium adsorption isotherm was well described by the Langmuir and Freundlich equations. The XMPC reached adsorption equilibrium at 303 K after approximately 120 min, and the removal rate of Cd(II) ions was 307 mg/g. The composite material can be reused many times and is easily magnetically separated from the solution. This makes the XMPC a promising candidate for widespread application in sewage treatment systems for the removal of heavy metals.

## 1. Introduction

As the global population increases and the world becomes more industrialized, water pollution has become a rapidly escalating global environmental problem that threatens the lives of the planet’s inhabitants. Water pollution originates from a variety of sources, and the availability of safe drinking water is of global concern, with the number of detected pollutants increasing, including inorganic anions, dyes, oil spills, and heavy metals. Heavy metals enter the environment from various sources, such as combustion, wastewater discharge, and production sites [1,2,3]. Heavy metal ions are water soluble and non-biodegradable and have many environmental, economic, and public health impacts [4,5]. Heavy metals are metals with densities greater than 5 g/cm^3^ and are usually present as trace elements. From an environmental perspective, the heavy metals of most concern are toxic heavy metals such as mercury (Hg), lead (Pb), cadmium (Cd), chromium (Cr), cobalt (Co), nickel (Ni), and copper (Cu) [6]. Excessive lead ion content can affect children’s intelligence, including their speaking ability, memory, and concentration. The safe limit for lead ions in drinking water is 100 µg/L, with the highest acceptable level being 50 µg/L [7,8]. The European Union classifies and regulates cadmium ions as highly hazardous toxic substances and carcinogens, limits the amount of cadmium discharge into lakes, rivers, dumps, and farmland, and prohibits pesticides that contain cadmium ions. The allowable concentration of cadmium ions in drinking water is 10 ppb, and that in food coloring must be less than 15 ppm [9,10]. These heavy metals are discharged into the environment in many forms, of which ionic heavy metal pollution in water is the most common [11]. Even when the concentration of heavy metal ions in water may be low, heavy metals accumulate in the human environment and cause various diseases and disorders [12].

The main treatment methods for wastewater with heavy metal contamination are ion exchange [13], adsorption [14], chemical precipitation [15], redox methods [16], solvent extraction [17], and membrane filtration (such as electrodialysis, nanofiltration, reverse osmosis, microfiltration, and ultrafiltration) [18]. Adsorption has the advantages of simplicity, high efficiency, low energy consumption, and renewability [19,20]. Adsorption is widely used for the recovery of heavy metals in sewage treatment [7,8]. Traditional adsorbent materials are usually made from acrylic acid, acrylamide, or their derivatives by chemical crosslinking polymerization [21]. Traditional adsorbent materials have relatively low specific surface areas and adsorption rates and cannot be easily recycled many times, all of which limit their application in practice [22].

Various adsorption materials, such as silica gel, activated carbon, resins, metal–organic frameworks, clays, and polymers have been used extensively to remove various pollutants from water bodies [23,24]. Membranes based on natural biopolymers are used in applications such as microfiltration, ultrafiltration, reverse osmosis, and nanofiltration membrane processes because of their low cost, availability, biodegradability, and natural origin [4]. The adsorbent chitosan is used extensively to treat water that has been polluted by harmful metal ions, antibiotic residues, and other pollutants, because of its high content of hydroxyl and amino functional groups and its biocompatibility, low toxicity, and biodegradability [25,26,27]. Chitosan has several shortcomings that limit its practical application: mechanical instability, sensitivity to pH, a challenging separation and recovery process, and low equilibrium capacity [28]. Several different modification methods have been studied to overcome these limiting factors [29,30]. The stability of chitosan in acidic solutions is increased by modifying it using crosslinking agents such as glyoxal, glutaraldehyde, and ethylenediamine, which usually act on the -NH_2_ or -OH of chitosan to change its chemical properties. Cross-linking can also improve the pore size distribution and adsorption/desorption performance [31].

To improve the mechanical and chemical properties of chitosan, polyvinyl alcohol (PVA) has been mixed with chitosan to form hydrogen bonds [32]. As a result, the hydrophobic side chains of the blend can be aggregated, and the intermolecular and intramolecular hydrogen bonds can interact [33]. Composites of chitosan and PVA have high adsorption capacity and are promising candidates for a broad range of applications in material development and wastewater treatment [34].

Chitosan and its derivatives are often prepared as gels, films, particles, and powders in practical applications [35]. Particulates are the most widely used form of adsorbent because their high specific surface areas result in excellent adsorption capacities [36]. However, the separation and recovery of heavy metals from particulate adsorbents are complex, cost-intensive, and time-consuming [37]. Magnetic separation technology can effectively separate magnetic materials from aqueous solutions and has the advantages of low operating cost, fast separation speed, and large processing capacity [38,39]. Magnetite (Fe_3_O_4_) is widely used as a magnetic material blend with chitosan because of its chemical stability, biocompatibility, favorable magnetic properties, and large surface area [25]. However, because of the van der Waals force between the nano-Fe_3_O_4_ particles and the magnetic force, accumulation is prone to occur. This reduces the specific surface area of the magnetic hydrogel, which reduces its ability to adsorb heavy metals [40].

Mesoporous SiO_2_ is widely used in adsorption and separation because of its uniform pore structure, large specific surface area, and good pore size characteristics [41]. Wrapping SiO_2_ in the outer layer of nano-Fe_3_O_4_ can effectively slow down the accumulation of nano-Fe_3_O_4_ and improve the adsorption capacity of magnetic mesoporous microspheres for heavy metals [42]. To further enhance the adsorption capacity of chitosan, various modifications with new functional groups such as ethylenediamine, thiourea, isatin, and xanthate have been extensively studied [43]. The modification of chitosan with xanthate can improve the interaction between chitosan and heavy metals in solution, thereby increasing the adsorption capacity [44].

In this work, a novel magnetic xanthate-modified adsorbent was designed and synthesized by surface modification of Fe_3_O_4_@SiO_2_/PVA/CS (XMPC) with the aim of improving the properties of chitosan, including its adsorption capacity, reusability, and mechanical stability. The synthesized XMPC adsorbents were characterized, and their adsorption properties for the removal of Pb(II), Cd(II), and Cu(II) from aqueous solutions under different experimental conditions were studied. Additionally, the adsorption/desorption performance of XMPC was also evaluated in batch experiments.

## 2. Experimental Method

### 2.1. Materials

Chitosan and PVA-1788 were from Sinopharm Chemical Reagent Co., Ltd. (Shanghai, China). Other reagents and chemicals were from Sigma Aldrich (St. Louis, MO, USA), Acros Organics (Geel, Belgium), and Fisher Scientific (Hampton, NH, USA), unless otherwise noted, and were used without further purification. All other chemicals were commercially available analytical-grade reagents. All solutions were prepared with deionized water.

### 2.2. Preparation of XMPC

Magnetic CS/PVA was prepared by a momentary gelation method. The chitosan–acetate solution was prepared by the dissolution of chitosan powder (6 g) in 150 mL of 2% (*v/v*) aqueous acetic acid at room temperature, and the PVA solution was prepared by the dissolution of 6 g of PVA powder in 150 mL of deionized water that was added to the chitosan–acetate solution. Fe_3_O_4_@SiO_2_ (1.2 g) was added into the mixture. The composite gel-forming solution was stirred continuously for 3 h at 30 °C until the Fe_3_O_4_@SiO_2_/PVA/CS (MPC) solution became a homogeneous magnetic gel solution. The gel-forming solution was dropped slowly into 1.0 M sodium hydroxide (NaOH), which resulted in the immediate formation of spherical hydrogel beads. The beads were gelled for 1 h and washed ten times with deionized water. Wet beads were dipped into glutaraldehyde solution (0.046 mL, 0.12 mmol) and stirred for 12 h at 30 °C to obtain crosslinked magnetic CS/PVA. After thorough washing with deionized water, the crosslinked magnetic PVA/CS beads were filtered and dried at 70 °C for 24 h. The PVA/CS beads (2 g) were treated with 100 mL of 14% NaOH solution and 1 mL of carbon disulfide (CS_2_). The mixture was stirred at room temperature for 24 h. The product was washed thoroughly with distilled water and dried at 70 °C for 24 h.

### 2.3. Analytical Methods

Fourier transform infrared spectroscopy (FTIR) measurements were performed on a Nicolet 6700 spectrometer equipped with an MCT detector. Thermogravimetric analysis (TGA) was undertaken with a NETZSCH STA 449C instrument, and measurements were performed from 25 °C to 700 °C, at a heating rate of 20 °C/min in N_2_. Differential scanning calorimetry (DSC) was performed on a NETZSCH DSC 200 PC unit from –50 °C to 300 °C, at a heating rate of 10 °C/min in N_2_. Raman spectra were obtained with a laser confocal microscope spectrometer (American Thermoelectric Corporation). The surface morphologies of the XMPC were visualized by SEM (SSX-550, Shimadzu, Kyoto, Japan). An X-ray diffraction (XRD) study of the samples was carried out on a Bruker D8 Focus +X-ray diffractometer operating at 30 kV and 20 mA with a copper target (l = 1.54 Å) and at a scanning rate of 1° min^−1^. The surface areas were determined by Brunauer–Emmett–Teller (BET) analysis (AUTOSORBiQ2, Quantachrome, Boynton Beach, FL, USA). Metal ion concentrations were determined by atomic absorption spectroscopy (SSX-550, Shimadzu, Kyoto, Japan).

## 3. Results and Discussion

### 3.1. Preparation and Characterization

The procedure used to synthesize the XMPC is illustrated in Figure 1. First, chitosan and PVA were dissolved in an acetic acid solution, in which the PVA had good dispersion stabilization. Fe_3_O_4_@SiO_2_ was then added such that it dispersed evenly, and the MPC was prepared by crosslinking with glutaraldehyde. The MPC was modified by reaction with CS_2_ under alkaline conditions to obtain the xanthate-modified adsorbent XMPC.

The infrared spectra of unmodified magnetic PVA/CS (MPC) and xanthate-modified magnetic PVA/CS (XMPC) in the range of 500–4000 cm^−1^ are shown in Figure 1a. The primary characteristic bands were as follows: in-plane bending vibration of O–H corresponding to the peak at 1400 cm^−1^, stretching vibration of C–O–C corresponding to the peak at 1162 cm^−1^, and absorption peaks at 3415 cm^−1^ and 581 cm^−1^ caused by the stretching of Fe–O bonds, indicating the presence of Fe_3_O_4_ in magnetic chitosan [30]. The absorption peaks at 1110 cm^−1^ were caused by the Si–O–Si vibrations, which also confirmed the formation of silicon shells on the surface of the Fe_3_O_4_ [45,46,47,48]. After the modification, the intensity of the characteristic peaks weakened, and the peak characteristics of C=S and S–C–S appeared at 1214 cm^−1^, indicating the successful production of the thiol-based modified chitosan hydrogel [49,50]. The infrared spectrum of the XMPC indicated the presence of chitosan and Fe_3_O_4_@SiO_2_. However, the peak indicating carbon disulfide modification was not pronounced, and the results of further analysis by Raman spectroscopy are shown in Figure 1b. The Raman spectrum of the MPC had two prominent peaks near 1345 cm^−1^ (D peak) and 1549 cm^−1^ (G peak). The peak at 1345 cm^−1^ was the bending vibration peak of NH, and the peak at 1549 cm^−1^ was the stretching vibration peak of C=C. After modification with xanthate, the D peak and G peak of the XMPC were weaker, and a peak characteristic of the C–S single bond appeared at 2908 cm^−1^, indicating the success of xanthate modification. Based on the comparison with the peak data of the Joint Committee on Powder Diffraction Standards file, the diffraction patterns for Fe_3_O_4_ and SiO_2_ (Figure 1c) were found to have broad peaks at 30.17°, 35.08°, 43.00°, 56.92°, and 62.45°, corresponding to (2 2 0), (3 1 1), (4 0 0), (5 1 1), and (4 4 0), respectively [51]. The diffraction intensity of the prepared XMPC decreased, indicating a decline in crystallinity. This decline was due to intramolecular hydrogen bonding with the polymer and the intermolecular hydrogen bonding interaction of the amino and hydroxyl groups to chitosan and PVA, respectively. This result indicated that Fe_3_O_4_@SiO_2_ nanoparticles were successfully introduced into the PVA/CS material.

The thermal characteristics of the XMPC were determined by its thermogravimetric analysis (TG) and differential scanning calorimeter (DSC) curves, and the results are shown in Figure 2a,b. In the TG and DTG curves, slight weight loss occurred below 150 °C, corresponding to the evaporation of absorbed and linked water in the material. The most significant weight loss occurred between 150 °C and 450 °C as a result of the structural degradation of CS and PVA. The final weight loss between 450 °C and 750 °C was attributed to charring due to the temperature exceeding the maximum degradation temperature of the material [52]. In the DSC curve, shown in Figure 2b, an exothermic peak was observed at approximately 101.2 °C, which may be due to the release of bound water in the XMPC. This was supported by the results of TGA. The glass transition temperatures (Tg) of PVA and CS were not found in the DSC curve, which also indicated that the new material, the XMPC, was formed successfully.

The magnetization curve of the hydrogel XMPC and its magnetic responsive behavior are shown in Figure 2c. The magnetic hysteresis loop with an S-like shape was symmetrical about the origin without hysteresis, remanence, or coercivity, implying that the material was superparamagnetic. The saturation magnetization of the XMPC was 18.79 emu/g, which is sufficient for magnetic separation in a solution. As the inset of Figure 2c shows, when a magnet was placed close to the glass bottle, the adsorbent material quickly gathered on the side of the glass bottle close to the magnet, indicating that the adsorbent material had favorable magnetic response characteristics [53].

Figure 2d shows the N_2_ adsorption–desorption isotherm of the XMPC, and the inset shows its Barrett–Joyner–Halenda (BJH) pore size distribution. The Brunauer–Emmett–Teller (BET) isotherm of the XMPC was characteristic of type IV adsorption–desorption behavior. When the relative pressure (P/P_0_) was within the range of 0.70 to 0.95, the desorption isotherm exhibited hysteresis typical of a mesoporous structure. The specific surface area of the material calculated from the N_2_ adsorption–desorption isotherm using the BET equation was found to be 3.65 m^2^/g, and the total pore volume was found to be 0.016 cm^3^/g. The BJH pore size distribution of the XMPC was between 0.5 nm and 5 nm and was mainly concentrated in a single peak. These results indicated that the material had a low specific surface area [54]. The microstructures of the XMPC and MPC were characterized by SEM (Figure 3). Their morphologies were similar, and the modified surface chemical groups did not change their original morphologies. The XMPC had a loose structure with interconnected pores, which enhanced the adsorption capacity, providing more adsorption sites on the surface and inside the adsorbent.

The XMPC was tested in deionized water and acidic and primary media for 10 d; the related data are shown in Appendix A. There was no significant change in mass after 10 d (˂1.0%). This may be due to the crosslinking of glutaraldehyde and the hydrogen bond interactions of PVA, CS, and Fe_3_O_4_@SiO_2_. This indicated that the modified material had enhanced durability in acidic and primary environments.

### 3.2. Evaluation of Adsorption and Desorption Performance

The adsorption uptake of heavy metal ions in water by the magnetic composite is shown as a function of time in Figure 4. The XMPC had a good adsorption effect on heavy metal ions in wastewater and is thus an excellent choice for treating wastewater. In the initial stage of adsorption, because of the higher concentration of the heavy metal ion solution, there were more adsorption groups on the hydrogel, and the adsorption rate of the heavy metal ions by the hydrogel was also faster. However, as contact time increased, the adsorption groups on the hydrogel were gradually filled with heavy metal ions, and the forces between the heavy metal ions on the hydrogel and the heavy metal ions in the water became stronger. After approximately 2 h, the adsorption capacity reached saturation, and the adsorption process became stable. The adsorption capacities of the XMPC for Cu(II), Pb(II), and Cd(II) ions were 67 mg/g, 100 mg/g, and 307 mg/g, respectively. The adsorption capacities of MPC for the Cu(II), Pb(II), and Cd(II) ions under the same conditions were 51 mg/g, 82 mg/g, and 220 mg/g, respectively. Chitosan modification with xanthate resulted in an improved interaction between the chitosan and heavy metals in solution, which increased the adsorption capacity. Related adsorbents were compared with the current adsorbent (XMPC), and the results are summarized in Table 1.

To understand the adsorption process, the experimental data were modeled using pseudo-first-order, pseudo-second-order kinetic, and intra-particle diffusion models [55,56,57]. The fit equations are shown in the Appendix A. Table 2 shows that quasi-first-order kinetics cannot accurately describe the mechanism of adsorption of heavy metal ions in solution by the composite material. There was a strong linear relationship between t/Qt and t. The quasi-second-order kinetic equation more accurately described the adsorption mechanism of the hydrogel, indicating that the process of the adsorption of heavy metal ions by the composite material was chemical adsorption. At the beginning of the adsorption process, there were many vacancies on the surface of the hydrogel that could be occupied by heavy metal ions. As the adsorption time increased, these vacancies on the hydrogel were gradually occupied. A concentration difference developed between the surface and interior of the hydrogel, leading to further adsorption of heavy metal ions, with the adsorption rate being affected by the intra-particle diffusion rate. Based on the above results, a mechanism for Cd(II) removal was proposed, as shown in Figure 5.

The influence of temperature on the adsorption capacity of Pb(II) Cu(II), and Cd(II) ions by the XMPC was studied at 293 K, 303 K, and 313 K (Figure 6). The adsorption isotherms were fit using the Langmuir and Freundlich models (Appendix A) [71,72]. The fitting equations are provided in the Appendix A. The standard Gibbs free energy change (ΔG°), standard enthalpy change (ΔH°), and standard entropy change (ΔS°) as functions of temperature were determined using the equations in the Appendix A. The calculated thermodynamic parameters are presented in Table 3 and Table 4.

As the concentration of metal ions increased, the adsorption groups of the composite material more fully contacted the ions, thereby increasing the adsorption capacity. This indicated that the concentration of the heavy metal ion solution strongly affected the adsorption capacity of the hydrogel. Figure 6 shows that adsorption by the composite material increased with the increase in temperature, which may occur because the increased temperature promoted irregular movement of the hydrogel and heavy metal ions. When the temperature exceeded 313 K, the adsorption capacity decreased slightly.

Appendix A shows that the correlation coefficient for the Freundlich model was higher than that of the Langmuir model, which suggests that the adsorption process was bilayer adsorption. Larger values of K_0_ indicated a faster adsorption rate. As the temperature increased, the adsorption rate gradually increased until the temperature reached 313 K. At this temperature, the high speed of molecular motion inhibited adsorption by the magnetic chitosan hydrogel. The chemical reaction process by which the hydrogel adsorbs heavy metal ions was spontaneous. The heat of chemical reaction in the adsorption process was 6.56 KJ/mol for Pb^2+^, 1.23 KJ/mol for Cu^2+^, and 0.44 KJ/mol for Cd^2+^. The value of △H was positive, indicating that the adsorption process was an endothermic reaction, so increasing the temperature would increase the adsorption rate [73,74]. If △S is negative, it means that the system’s entropy is reduced, which means that the hydrogel has absorbed the heavy metal ions in the water. This reduces the number of molecules that move randomly in the water.

### 3.3. Selective Adsorption Behaviors of Heavy Metals

The coexistence of multiple ions was studied at a temperature of 293 K and an adsorption time of 120 min. The adsorption performance and mutual influence of the XMPC on various heavy metal ions is shown in Table 5. In the single-component adsorption system, the modified composite material had better adsorption performance for heavy metal ions, especially Cd(II), with an adsorption capacity of 300 mg/g. In the multi-component system, that is when two or five ions compete for adsorption sites, the adsorption capacity for various ions was significantly reduced, especially in the mixed solution of five ions. Compared with the adsorption capacity of Cd(II) in the single-component system, the adsorption capacity was reduced by 80%. A larger distribution coefficient of the heavy metal ions in the multi-component system indicated that the ions had stronger affinity. The affinity of Cd(II) was strongest, as was the competitiveness of the active sites on the magnetic hydrogel. The smaller the value of the selectivity coefficient, the better the selectivity of the magnetic hydrogel for the measured ion was. The number of adsorption sites on the modified magnetic chitosan composite hydrogel was limited could overlap easily with the active sites on the hydrogel. When the hydrogel preferentially adsorbed the most competitive Cd(II), the number of active sites occupied by other ions decreased, leading to different decreases in adsorption capacity for each different type of heavy metal ion. This experiment showed that during competitive adsorption, the affinity and selective adsorption of Cd(II) were both optimal.

### 3.4. Desorption and Regeneration

Desorption is essential to enable the reuse of the adsorbent and recover metal ions. Desorption measurements were also used to elucidate the adsorption process. The exhausted adsorbent was eluted using HCl (pH = 1). Nine consecutive re-adsorption/desorption cycles to evaluate the adsorption efficiency of metal ions on the XMPC were conducted under the same conditions, and the results are shown in Figure 7 and Appendix A. When the magnetic hydrogel adsorbed three heavy metal ions for the fourth time, the adsorption capacity was significantly reduced. As the magnetic hydrogel adsorption times increased, the ion adsorption capacity of the hydrogel decreased significantly. The curve in Figure 7c shows that the removal rate decreased as the number of cycles increased. Hydrochloric acid may destroy the structure of the magnetic hydrogel, thus causing the desorption rate to drop abruptly, and thereby affecting the overall adsorption performance of the magnetic hydrogel. After five cycles of adsorption and desorption, the adsorption capacity was still higher than 50%, which showed that the thiol-modified magnetic hydrogel could be regenerated and reused after desorption.

## 4. Conclusions

Magnetic hydrogel beads of the XMPC were prepared, and their adsorption of Pd(II), Cu(II), and Cd(II) ions was studied experimentally. The XMPC reached adsorption equilibrium after adsorption for about 120 min at 303 K, and the removal rates of Pd(II), Cu(II), and Cd(II) ions were 67 mg/g, 100 mg/g, and 307 mg/g, respectively. The adsorption of metal ions by the XMPC was a spontaneous and endothermic process and was well described by the quasi-secondary kinetic model and the Langmuir adsorption isotherm. The adsorbent was stable below 150 °C and had good magnetic properties. The adsorbent was found to be stable in deionized water, acidic, and basic media for at least 10 d. After five cycles of use, the removal rate of metal ions by the magnetic hydrogel beads remained above 50%. The results of this study showed that XMPC has not only the adsorption capacity of CS for metal ions, but also the magnetic properties of Fe_3_O_4_ and demonstrated the ease and speed of solid–liquid separation after adsorption by the XMPC.

## Data Availability

Not applicable.

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
