# Peer review of "Xanthate-Modified Magnetic Fe3O4@SiO2-Based Polyvinyl Alcohol/Chitosan Composite Material for Efficient Removal of Heavy Metal Ions from Water"

_polymers, 2022, doi:10.3390/polym14061107_

Round 1
Reviewer 1 Report
Detailed comments:
- The English of the text should be checked
- The novelty of manuscript should be highlighted moreAt paragraphs write at lines 27-33, authors must include more information about: the source of heavy metals, the level of toxicity when heavy metals become toxic, the forms of heavy metals discharged into the environment, the concentration of heavy metal ions in water, the effects of metal ions on human health, but also on living organisms in water
- At lines 34-36, for each treatment methods mentioned must be indicated the Reference. Also, for the membrane filtration must be indicated example and, must be included membrane processes (e.g. electrodialysis, reverse electrodialysis, reverse-osmosis, ultrafiltration, nanofiltration, microfiltration)
- The authors must be included new, relevant and more information about other adsorbent materials (e.g. membranes, resins (Purolite, andHypersol-Macronet MN500), montmorillonite). Also, must be included more advantages and disadvantage of adsorption in comparison with other membrane filtration and membrane The following references can be included in the Introduction part to improve the quality of manuscript, because they provide relevant information:
- Effective removal of heavy metal ions Cd2+, Zn2+, Pb2+, Cu2+ from aqueous solution by polymer-modified magnetic nanoparticles. J. Hazard. Mater. 2012, 211–212, 366–372.
- Study of Polyvinyl Alcohol-SiO2 Nanoparticles Polymeric Membrane in Wastewater Treatment Containing Zinc Ions, Polymers, 2021, 13(11), 1875
- Removal of Cu2+, Cd2+ and Ni2+ ions from aqueous solution using a novel chitosan/polyvinyl alcohol adsorptive membrane. Carbohydr. Polym. 2019, 210, 264–273.
- Biopolymeric Membrane Enriched with Chitosan and Silver for Metallic Ions Removal, Polymers 2020, 12, 1792
- Preparation and characterization of membranes obtained from blends of acrylonitrile copolymers with poly(vinyl alcohol), Appl. Polym. Sci. 131 (2014),
- For the procedure used to synthesize XMPC must be indicate more information: operation conditions, concentration and amount for each chemical, reagent used, temperature, etc.
- Lines 95, 96: “The MPC was modified by reaction with CS2 under alkaline conditions to obtain the xanthate-modified adsorbent XMPC” – what represent CS2, concentration, amount; also, what means under alkaline conditions? More information must be includedThe principle of operation of the process shown in the Figure 1 must be detailed in the text of the manuscript. Moreover, any abbreviation, notation must be made explicit, what does it represent?
- Figures 1, 4, 5, 6 must be done on a large scale
- SEM image of MPC must be included. Also, the comparison between MPC and XMPC must be done
- For all characterization method and tests must be indicated equipment, operational conditions, temperature
- Lines 161-163; authors write: “When the pH of the solution is lower than 5.5, chitosan dissolves, which limits its range of applications. XMPC was tested in deionized water and acidic and primary media for 10 days” – you must indicate which solution was used;why at the pH value of 5.5 the range of applications are limited, what are those applications;fordeionized water and acidic and primary media must be indicate the amount and concentration of each media; also, explain why 10 days were tested and not less or more days
- Lines 189-197, authors write: “The quasi-second-order kinetic equation more accurately described the adsorption mechanism of the hydrogel, indicating that the process of adsorption of heavy metal ions by the composite material is chemical adsorption. At the beginning of the adsorption process, there are many vacancies on the surface of the hydrogel that can be occupied by heavy metal ions. As the adsorption time increases, these vacancies on the hydrogel are gradually occupied. A concentration difference develops between the surface and interior of the hydrogel, leading to further adsorption of heavy metal ions, with the adsorption rate being affected by the intra-particle diffusion rate.” - a schematic mechanism describing the adsorption process must be indicated and included (reactions, interactions etc.)
- Lines 254-255, authors write: “The exhausted adsorbent was eluted using HCl (pH = 1).” – explain why was use the HCl
- Comparison between the obtained results and measured in this study with other reported studies should be done and included for more clarity (indicate values not just number of reference).
- Correct the References using the guide of Journal. More Conclusions must be included with the best results, values obtained
- Reference 6 is the same with Reference 22
- The manuscript contains same Reference of the authors [28, 30, 40]. What is the difference between the materials indicated in this manuscript and those already published by the authors?
- Same Reference are very old [3, 13, 15, 31, 34, 41, 42, 43, 44, 45 (years 1918, 1981? – these can be replaced with books)]. Please eliminate same of theme because are very many and replaced with other new. The manuscript must contain the relevant information to be attractive for readers (researchers), because science has advanced, and the information indicated in the manuscript is no longer valid. This part should include observed information, noted in the last 10 years.
Author Response
Response to Review #1
1. We have revised the writing.
2-4. We are very grateful for the reviewer’s pertinent suggestions to help improve the manuscript. We added the sentence of “As the global population increases and the world becomes more industrialized, water pollution has become a rapidly escalating global environmental problem that threatens the lives of the planet's inhabitants. Water pollution originates from a variety of sources, and the availability of safe drinking water is of global concern, with the number of detected pollutants increasing, including inorganic anions, dyes, oil spills, and heavy metals. Heavy metals enter the environment from various sources, such as combustion, wastewater discharge, and production sites[1-3]. Heavy metal ions are water-soluble and non-biodegradable, and have many environmental, economic, and public health impacts[4,5].”,” Even when the concentration of heavy metal ions in water may be low, heavy metals accumulate in the human environment and cause various diseases and disorders.”,” Excessive lead ion content can affect children's intelligence, including their speaking ability, memory, and concentration. The safe limit for lead ions in drinking water is 100 micrograms/liter, with the highest acceptable level being 50 micrograms/liter [7,8]. The European Union classifies and regulates cadmium ions as highly hazardous toxic substances and carcinogens; limits the amount of cadmium discharge into lakes, rivers, dumps and farmland; and prohibits pesticides that contain cadmium ions. The allowable concentration of cadmium ions in drinking water is 10 ppb, and that in food coloring must be less than 15 ppm[9,10].”,” The main treatment methods for wastewater with heavy metal contamination are ion exchange[13], adsorption[14], chemical precipitation[15], redox methods[16], solvent extraction[17], and membrane filtration (such as electrodialysis, nanofiltration, reverse osmosis, microfiltration, and ultrafiltration) [18].”,“Various adsorption materials, such as silica gel, activated carbon, resins, metal organic frameworks, clays, and polymers have been used extensively to remove various pollutants from water bodies [23,24]. Membranes based on natural biopolymers are used in applications such as microfiltration, ultrafiltration, reverse osmosis, and nanofiltration membrane processes because of their low cost, availability, biodegradability, and natural origin [4].” in the Introduction section (Page 1, 2). At the same time, relevant important documents are cited.
5-6. Thank you for your comments, we have moved parts of Supplementary Materials to the main text, including the preparation of the materials.
7. We provide the original image, which will be improved in the later typesetting.
8. We fully agree with the reviewer’s comments, we added SEM image of the MPC in Figure 3. We also revised the sentence of “The microstructures of XMPC and MPC were characterized by SEM (Fig. 3). Their morphologies are similar, and the modified surface chemical groups do not change their original morphologies. XMPC has a loose structure with interconnected pores, which enhances the adsorption capacity providing more adsorption sites on the surface and inside of the adsorbent.” in Page 7.
9. We added 2. Experimental Method in the main text and some additional test conditions in the Supplementary Materials.
10. Pure chitosan is limited by pH of solution because chitosan dissolves when the pH ranges below 5.5. To avoid misunderstandings, we deleted the sentence of “When the pH of the solution is lower than 5.5, chitosan dissolves, which limits its range of applications” .We have revised the sentence as “To evaluate the durability of XMPC, deionized water (pH = 7), acidic (pH = 3, aqueous hydrochloric acid), and basic media (pH = 11, aqueous sodium hydroxide) were used for the swelling test. 1 g of sample immersed in 200 mL deionized water, acidic, and basic media for 10 days, respectively. Then the swollen samples were weighted immediately after removing excess water.” in Supplementary Materials. The reported swelling experiments of materials include 24h (https://doi.org/10.1016/j.carbpol.2019.01.074), 48h (https://doi.org/10.1016/j.chemosphere.2019.02.199), and 10 days (https://doi.org/10.1016/j.jtice.2017.06.009). We want to test the stability of the material and choose 10 days.
11. We added Fig. 5. The proposed adsorption mechanism using XMPC capture Cd(II) .
Figure 5. The proposed adsorption mechanism using XMPC capture Cd(II).
- The material can be desorbed under strong acid conditions, and both hydrochloric acid and nitric acid have been used in the reported literature (https://doi.org/10.1016/j.jhazmat.2011.12.013,https://doi.org/10.1016/j.cej.2018.10.001). Nitric acid has strong oxidizing properties and can generate NO, so we choose to use hydrochloric acid.
- We added Table 2 Various chitosan-based adsorbents designed for removing of Cu(II), Pb(II), and Cd(II) ions from aqueous solutions.
adsorbent |
Qm (mg/g) |
Conditions |
Ref. |
|||
Cu(II) |
Pd(II) |
Cd(II) |
pH |
T (oC) |
||
Amidoxime-Functionalized CS |
190.7 |
- |
- |
5 |
25 |
[12] |
MMT/CS |
17.2 |
- |
- |
9 |
45 |
[53] |
Magnetic bentonite/Carboxymethyl CS/SA hydrogel beads |
56.79 |
- |
- |
5 |
30 |
[20] |
Zeolite X/CS hybrid microspheres |
152.0 |
- |
- |
5.5 |
25 |
[2] |
Silica/CS composite |
870 |
316 |
- |
5 |
30 |
[69] |
CNTs-CHO-CS composite |
115.84 |
- |
- |
7 |
25 |
[14] |
CS-g-MA composite |
312.4 |
- |
- |
6 |
25 |
[16] |
CS/PVA/PEI membrane |
86.08 |
- |
112.13 |
6 |
25 |
[70] |
CS-g-PAA/APT |
303.03 |
- |
- |
5.5 |
30 |
[71] |
CS-MMT hydrogel |
132.74 |
- |
- |
5 |
20 |
[72] |
DTPA-CS/PEO nanofibers |
177.0 |
142.0 |
- |
5 |
20 |
[17] |
CS/TEOS/APTES nanofiber |
640.5 |
575.5 |
- |
6, 5.5 |
45 |
[73] |
TEPA/CS/CoFe2O4 composite |
168.067 |
228.311 |
- |
5 |
30 |
[74] |
Alginate/Melamine/CS aerogel |
- |
1331.6 |
- |
5.5 |
25 |
[75] |
polydopamine modified CS aerogels |
- |
441.2 |
- |
5 |
45 |
[76] |
Magnetic-CS–PAA nanocomposite |
- |
204.9 |
- |
6 |
35 |
[77] |
PEI-grafted magnetic CS microspheres |
- |
134.9 |
- |
|
|
[78] |
CS-pectin gel beads |
169.4 |
266.5 |
177.6 |
4-9 |
50 |
[79] |
Crosslinked carboxylated CS/ carboxylated nanocellulose hydrogel beads |
- |
334.9 |
- |
4 |
35 |
[80] |
Hydroxyapatite/CS composites |
- |
132.1 |
81.1 |
6 |
25 |
[63] |
CS-PVA nanofibers |
- |
266.12 |
148.79 |
6, 8 |
25 |
[64] |
MnO2 /CS nanoparticles |
- |
126.1 |
- |
4 |
50 |
[65] |
sodium tripolyphosphate cross-linked CS beads |
- |
- |
99.8 |
7 |
55 |
[66] |
Vermiculite blended CS |
- |
- |
169.0 |
5.5 |
30 |
[67] |
Thiourea-modified magnetic ion-imprinted CS/TiO2 |
- |
- |
256.41 |
7 |
25 |
[9] |
CS/PVA/PEI membrane |
86.08 |
- |
112.13 |
6 |
25 |
[10] |
Cobalt ferrite@SiO 2 -CS/EDTA composite |
- |
- |
127.79 |
6 |
25 |
[81] |
CS@NZVI |
- |
- |
142.80 |
7 |
25 |
[82] |
CS-VMT composite |
- |
166.67 |
55.48 |
4 |
30 |
[83] |
XMPC |
100 |
67 |
307 |
5.5 |
30 |
This work |
14,15,17. Thank you for your comments, we have revised the References.
- [28] Wang, Q.; Tian, Y.; Kong, L. et al. unveiled a novel polyethyleneimine (PEI) functionalized chitosan (CS) aerogel (PCA) with three-dimensional (3D) porous network structure and superior reversible compression property and investigated its adsorption behavior toward Cr(VI). [30] Chen, Y.; Wang, J. prepared and characterized a magnetic chitosan nanoparticle and applied it to remove Cu(II). [40] Zeng, H.; Wang, L.; Zhang, D. et al. reported a novel polymer-based adsorbent of hyperbranched polyethylenimine functionalized carboxymethyl chitosan semi-interpenetrating network composite (HPFC) was fabricated through a facile one-step crosslinking reaction. HPFC can adsorb up to 1594 mg/g Hg(II) ions from aqueous solution. They are all composite materials based on chitosan, but the modification methods are different.

Reviewer 2 Report
"Xanthate-modified magnetic Fe3O4@SiO2-based polyvinyl alcohol/chitosan composite material for efficient removal of heavy metal ions from water" describes the synthesis of a magnetic xanthate-modified magnetic polyvinyl alcohol and chitosan composite (XMPC) for the efficient removal and recovery of heavy metal ions from aqueous solution which is a good topic and falls in the topic of the journal, however, there are some issues to be addressed. the comments are listed below:
1- chitosan main properties should be discussed in introduction (Carbohydrate Polymers 274 (2021): 118671, Journal of hazardous materials 408 (2021): 124889., Arabian Journal of Chemistry 15.2 (2022): 103543, Arabian Journal of Chemistry (2022): 103743)
2- the adsorption performance of the unmodified MPC composite must be examined to clarify the role of xanthante modifications.
3- Fig. 1c should be improved.
4- In FTIR, check the band related to Si-O-Si.
5- the prepared XMPC composite is the form of beads which is not shown by either SEM or digital photos which in necessary for reliability. Furthermore, SEM image of MPC befor modification should be added and discussed in the article.
6- Supplementry information should be cited in the main text specially the preparation part.
7- authors stated that "The metal ion adsorption capacity increased 203
with temperature, indicating that the process was endothermic", however, for Cu(II) the adsorption capacity increased by increasing temperature then decreased with further increase in temperature. please check this point.
8- In Table 3, the unit of ΔS should be in prackets.
9- In Fig. 4, 5 & 6 the symbols a, b and c must be cited in the figure caption.
10- English language should be checked throughout the Ms.
Author Response
Response to Review #2
- We are very grateful for the reviewer’s pertinent suggestions to help improve the manuscript. We added the sentence of “Various adsorption materials, such as silica gel, activated carbon, resins, metal organic frameworks, clays, and polymers have been used extensively to remove various pollutants from water bodies [23,24].”, “The adsorbent chitosan is used extensively to treat water that has been polluted by harmful metal ions, antibiotic residues, and other pollutants, because of its high content of hydroxyl and amino functional groups and its biocompatibility, low toxicity, and biodegradability [25-27]” in the Introduction section (Page 2). At the same time, relevant important documents are cited.
- We fully agree with the reviewer’s comments, We added the sentence of “The adsorption capacities of MPC for Cu(II), Pb(II), and Cd(II) ions under the same conditions were 51 mg/g, 82 mg/g, and 220 mg/g, respectively. Chitosan modification with xanthate resulted in an improved interaction between the chitosan and heavy metals in solution, which increased the adsorption capacity.” in Page 8.
- We have improved Fig.1c.
- We have revised the sentence as “The absorption peaks at 1110 cm-1 are caused by the Si–O–Si vibrations, which also confirms the formation of silicon shells on the surface of the Fe3O4” in Page 5.
- We added SEM image of the MPC in Figure 3. We also revised the sentence of “The microstructures of XMPC and MPC were characterized by SEM (Fig. 3). Their morphologies are similar, and the modified surface chemical groups do not change their original morphologies. XMPC has a loose structure with interconnected pores, which enhances the adsorption capacity providing more adsorption sites on the surface and inside of the adsorbent.” in Page 7. The prepared XMPC composites are beads with a radius of about 1-3 nm. In the SEM test, we selected a bead to magnify the picture by 5000 times.
- We have moved parts of Supplementary Materials to the main text, including the preparation of the materials.
- We removed this sentence and explained it as follows “Fig. 6 shows that adsorption by the composite material increases with increasing temperature, which may be because the increased temperature promotes the irregular movement of hydrogel and heavy metal ions. Regardless, once the temperature reaches 313 K and above, the adsorption capacity decreases slightly.” in Page 10.
8, 9. Thank you for your comments, we have revised the relevant content
- It is our great apology for our unconscious mistake about writing. We have revised the writing.

Round 2
Reviewer 1 Report
Accept